# Assessment of Cross-Reactivity of Chimeric *Trypanosoma cruzi* Antigens with *Crithidia* sp. LVH-60A: Implications for Accurate Diagnostics

**DOI:** 10.3390/diagnostics13223470

**Published:** 2023-11-17

**Authors:** Emily F. Santos, Ramona T. Daltro, Carlos G. Regis-Silva, Tycha B. S. Pavan, Fabrícia A. de Oliveira, Ângela M. da Silva, Roque P. Almeida, Noilson L. S. Gonçalves, Daniel D. Sampaio, Faber N. Santos, Fabricio K. Marchini, Paola A. F. Celedon, Nilson I. T. Zanchin, Fred L. N. Santos

**Affiliations:** 1Advanced Public Health Laboratory, Gonçalo Moniz Institute, Oswaldo Cruz Foundation (Fiocruz-BA), Salvador 40296-710, BA, Brazil; emily.santos@fiocruz.br (E.F.S.); carlos.regis@fiocruz.br (C.G.R.-S.); tycha.pavan@fiocruz.br (T.B.S.P.); noilson.goncalves@fiocruz.br (N.L.S.G.); 2Integrated Translational Program in Chagas Disease from Fiocruz (Fio-Chagas), Oswaldo Cruz Foundation (Fiocruz-RJ), Rio de Janeiro 21040-360, RJ, Brazil; ramona.daltro@fiocruz.br (R.T.D.); nilson.zanchin@fiocruz.br (N.I.T.Z.); 3Immunology and Molecular Biology Laboratory, University Hospital/EBSERH, Federal University of Sergipe, Aracaju 49060-676, SE, Brazil; fa_alvisi@hotmail.com (F.A.d.O.); angela.silva910@gmail.com (Â.M.d.S.); roquepachecoalmeida@gmail.com (R.P.A.); 4Department of Medicine, University Hospital (BSERH), Federal University of Sergipe, Aracaju 49060-676, SE, Brazil; 5Brazil’s Family Health Strategy, Municipal Health Department, Tremedal City Hall, Tremedal 45170-000, BA, Brazil; diassampaio@gmail.com; 6Faculty of Medicine, Santo Agostinho College (FASA), Education Technology Healthcare, Vitória da Conquista 45028-100, BA, Brazil; dr.faber@hotmail.com; 7Molecular Biology Institute of Paraná, Curitiba 81350-010, PR, Brazil; fabricio.marchini@fiocruz.br; 8Laboratory for Applied Science and Technology in Health, Carlos Chagas Institute (ICC), Oswaldo Cruz Foundation (Fiocruz-PR), Curitiba 81350-010, PR, Brazil; 9Laboratory of Molecular and Systems Biology of Trypanosomatids, Carlos Chagas Institute, Oswaldo Cruz Foundation (Fiocruz-PR), Curitiba 81350-010, PR, Brazil; paola.fiorani@fiocruz.br; 10Structural Biology and Protein Engineering Laboratory, Carlos Chagas Institute, Oswaldo Cruz Foundation (Fiocruz-PR), Curitiba 81350-010, PR, Brazil

**Keywords:** Chagas disease, trypanosomatids, *Crithidia* sp. LVH-60A, diagnosis, cross-reaction, recombinant antigen

## Abstract

This study focuses on developing accurate immunoassays for diagnosing Chagas disease (CD), a challenging task due to antigenic similarities between *Trypanosoma cruzi* and other parasites, leading to cross-reactivity. To address this challenge, chimeric recombinant *T. cruzi* antigens (IBMP-8.1, IBMP-8.2, IBMP-8.3, and IBMP-8.4) were synthesized to enhance specificity and reduce cross-reactivity in tests. While these antigens showed minimal cross-reactivity with leishmaniasis, their performance with other trypanosomatid infections was unclear. This study aimed to assess the diagnostic potential of these IBMP antigens for detecting CD in patients with *Crithidia* sp. LVH-60A, a parasite linked to visceral leishmaniasis-like symptoms in Brazil. This study involved seven *Crithidia* sp. LVH-60A patients and three *Leishmania infantum* patients. The results indicated that these IBMP antigens displayed 100% sensitivity, with specificity ranging from 87.5% to 100%, and accuracy values between 90% and 100%. No cross-reactivity was observed with *Crithidia* sp. LVH-60A, and only one *L. infantum*-positive sample showed limited cross-reactivity with IBMP-8.1. This study suggests that IBMP antigens offer promising diagnostic performance, with minimal cross-reactivity in regions where *T. cruzi* and other trypanosomatids are prevalent. However, further research with a larger number of *Crithidia* sp. LVH-60A-positive samples is needed to comprehensively evaluate antigen cross-reactivity.

## 1. Introduction

Chagas disease (CD) is a neglected tropical disease caused by *Trypanosoma cruzi*, a hemoflagellate protozoan of the family Trypanosomatidae [1]. It poses a significant public health challenge in 21 Central and South American countries, with an estimated 5.7 million people infected and an annual count of 10,000 CD-related deaths [2]. The World Health Organization (WHO) estimates that approximately 70 million people live in areas at risk of infection [3]. In recent decades, changing migration patterns have caused a global shift in the prevalence of this disease, turning it into a significant public health concern in Europe, North America, and the Western Pacific [4,5].

Transmission primarily occurs through the feces of bloodsucking triatomines (Hemiptera: Reduviidae), commonly known as kissing bugs. Secondary routes of transmission include the consumption of contaminated beverages and food [6], mother-to-child transmission during pregnancy [7], whole-blood or blood-derivative transfusion [8], tissue and organ transplantation [9], and exposure through laboratory accidents [10]. CD is clinically categorized into two phases: acute and chronic. The acute phase, lasting 2–3 months, exhibits nonspecific symptoms including fever, headache, fatigue, muscle pain, and lymphadenopathy [11]. Microscopic detection of motile forms of *T. cruzi* in the blood is utilized for diagnosis during this stage. With lifelong chronic infection, the parasite can invade the heart and the digestive and nervous systems in 30% of *T. cruzi*-infected individuals, leading to chronic health complications such as heart disease and digestive disorders [12]. Specific anti-*T. cruzi* antibodies (IgG) are detected during this stage using antibody/antigen-based methods due to the low and intermittent parasitemia [13,14].

Several serological methods are available for diagnosing chronic CD, including enzyme-linked immunosorbent assay (ELISA) [13], indirect hemagglutination [15], indirect immunofluorescence [15], Western blot [16], lateral flow assays [17], flow cytometry [18], and optical reader-based methods such as liquid microarray or immunosensors [19]. The performance of these assays relies on the antigenic preparation used to detect anti-*T. cruzi* antibodies [13]. Conventional assays employ fractionated *T. cruzi* lysates or whole-cell epimastigote homogenates, resulting in a complex antigen mixture with an unknown and variable composition [13]. Despite their high sensitivity, these tests have limitations that hinder their utilization, including difficulties in standardizing the methods, specificity problems, and cross-reactivity with *Leishmania* spp. and *Trypanosoma rangeli* [20,21,22], which share significant genetic similarity with *T. cruzi* [23,24]. Unconventional assays utilizing recombinant antigens aim to overcome interference with other components that arises when antigens are extracted from whole parasites. However, cross-reactivity with *Leishmania* spp. has been consistently observed with recombinant antigens [13,20,25,26,27]. As a result, commercial tests employing recombinant antigen preparations are prone to yielding false-positive results in individuals exposed to *Leishmania* spp., especially in regions where both infections coexist.

Despite the availability of various methodologies, technical and operational constraints have led to irregular performance in serological assays. These constraints occur due to the significant genetic and phenotypic diversity within *T. cruzi* [8], which affects the choice of antigens used in immunoassays [9], variations in disease prevalence [10,11], and variable immune responses in individuals infected with *T. cruzi* [12]. As a result, the World Health Organization (WHO) recommends the simultaneous use of two different serological tests for CD diagnosis. The primary challenge in diagnosing chronic CD lies in the absence of effective tools for large-scale screening and point-of-care diagnosis across various epidemiological scenarios.

Furthermore, standard diagnostic protocols are difficult to implement outside of major urban centers and come with several constraints, including the need for highly trained personnel, specific equipment, refrigerated storage, and multiple visits to healthcare facilities. This underscores the importance of developing new diagnostic tools tailored to the needs of affected populations and the realities of healthcare systems based on primary care. Such tools should be user-friendly and have the potential to significantly enhance the detection of CD carriers.

One strategy to address these limitations, including the inconsistent performance of serological assays in different settings, involves the use of recombinant chimeric antigens. These antigens consist of repetitive and conserved amino acid sequences from epitopes found in several antigenic proteins of the parasite [14,15].

Recent advancements in DNA recombination technology have enabled the synthesis of chimeric proteins comprising multiple antigenic peptides derived from different parasite proteins. These chimeric proteins offer the advantage of a lack of cross-reactivity with other pathogens [28,29,30,31]. In light of these developments, our research group has successfully produced and extensively investigated the diagnostic performance of four chimeric proteins (IBMP-8.1, IBMP-8.2, IBMP-8.3, and IBMP-8.4) for the detection of specific anti-*T. cruzi* antibodies [22,30,32,33]. In a study involving these chimeric *T. cruzi* antigens [22], minimal cross-reactivity was observed in both American cutaneous leishmaniasis and visceral leishmaniasis, demonstrating their suitability for regions where *T. cruzi* and *Leishmania* spp. coexist. In addition to *Leishmania* spp., other members of the Trypanosomatidae family, including *Crithidia*, *Herpetomonas*, *Blastocrithidia*, or *Leptomonas*, may also be present in *T. cruzi* endemic areas, thereby contributing to the diagnostic challenges associated with Chagas disease in these regions.

In 2011, a non-*Leishmania* parasite closely related to *Crithidia* was isolated from a 64-year-old man in the Brazilian state of Sergipe. Despite exhibiting clinical symptoms similar to visceral leishmaniasis, the standard treatment for leishmaniasis proved ineffective. Further examination revealed the presence of trypanosomatids in serum samples from individuals who tested positive for leishmaniasis using the rK39 test. Through genome sequencing, a new parasite belonging to the Trypanosomatidae family was identified [34]. This recently discovered parasite has been named *Crithidia* sp. LVH-60A (GeneBank number ASM3007807v1) [35] and is estimated to have infected over 150 individuals, resulting in two deaths in Sergipe [36]. The emergence of these unusual parasites presents challenges for serological diagnostics of CD. Therefore, the objective of this study was to evaluate the potential cross-reactivity of four chimeric IBMP antigens in combination with the statistical approach of latent class analysis (LCA) for detecting anti-*T. cruzi* IgG in serum samples obtained from individuals infected with *Crithidia* sp. LVH-60A.

## 2. Materials and Methods

### 2.1. IBMP Chimeric Antigen Preparation

*Trypanosoma cruzi* chimeric proteins (IBMP-8.1, IBMP-8.2, IBMP-8.3, IBMP-8.4) were produced and purified using established protocols [30]. Recombinant pET28a vector-transformed *Escherichia coli* BL21-Star (DE3) cells were cultured in lysogenic broth (LB) medium supplemented with 50 µg/mL kanamycin and 0.5 mM isopropyl β-D-1-thiogalactopyranoside (IPTG) at 37 °C for 4 h. Cell disruption was achieved using microfluidization (Microfluidizer M110L, Microfluidics, Westwood, MA, USA), followed by centrifugation to remove insoluble debris. Ion-exchange and liquid affinity chromatography techniques were applied to purify proteins from the supernatant. The recombinant expression of chimeras was confirmed using sodium dodecyl sulfate-polyacrylamide gel electrophoresis, and the concentrations of purified proteins were determined using a fluorimetric assay (Qubit 2.0, Invitrogen Technologies, Carlsbad, CA, USA).

### 2.2. Sample Collection

Human sera, previously collected, were obtained from the Biorepository of the Laboratory of Immunology and Molecular Biology at the Federal University of Sergipe (Aracaju, Sergipe, Brazil) [34]. A total of seven samples from individuals infected with *Crithidia* sp. LVH-60A and three samples from individuals infected with *L. infantum* were used to assess cross-reactivity against IBMP *T. cruzi* chimeric antigens.

### 2.3. Laboratory Assays (IBMP-Indirect ELISA)

Immunoassays were performed following established protocols [32]. Transparent 96-well flat-bottom microplates (UV-Star R Microplate, Greiner Bio-One, Kremsmünster, Austria) were coated with chimeric IBMP antigens at concentrations of 12.5 ng (IBMP-8.2) or 25 ng (IBMP-8.1, IBMP-8.3, and IBMP-8.4) per well. The coating was achieved by diluting the antigens in a 0.05 M carbonate/bicarbonate solution (pH 9.6). Next, the microplates were blocked with Well Champion reagent (Kem-En-Tec, Taastrup, Denmark) according to the manufacturer’s instructions. The serum samples were diluted 1:100 in 0.05 M phosphate-buffered saline (pH 7.2), and the microplates incubated at 37 °C for 60 min. Following incubation, the microplates were washed with PBS-0.05% Tween-20 (PBS-T; pH 7.4) to remove unbound antibodies. HRP-conjugated goat anti-human IgG (Bio-Manguinhos, FIOCRUZ, Rio de Janeiro, Brazil) was diluted 1:40,000 in PBS-T, and 100 μL of the diluted solution was added to each well. Incubation was carried out at 37 °C for 30 min. After another round of washing, 100 μL of TBM substrate (Kem-En-Tec Diagnostics A/S, Taastrup, Denmark) was added to detect immune complex formation. This was followed by a 10 min incubation at room temperature in the dark, and the colorimetric reactions were stopped by adding 50 μL of 0.3 M H_2_SO_4_ to the wells. Optical density was measured using a SPECTRAmax 340PC microplate reader with a 450 nm filter (Molecular Devices, San Jose, CA, USA), and background values were subtracted from the readings.

### 2.4. Latent Class Analysis (LCA) as a Reference Test

In the absence of a gold standard for diagnosing chronic CD, we employed latent class analysis (LCA) as a statistical approach for serological classification of *T. cruzi* infection. This well-established and validated approach [33] determined *T. cruzi* reactivity based on positive results from a minimum of two out of the four chimeric antigens, resulting in posterior probabilities ranging from 87.9% to 100%. Conversely, nonreactive status for *T. cruzi* was assigned when at least three out of the four chimeric antigen-based assays yielded negative results, with posterior probabilities ranging from 0% to 0.8%.

### 2.5. Data Analysis

The data were analyzed and visualized using Prism software (version 10; GraphPad, San Diego, CA, USA). Descriptive statistics were reported as geometric means ± standard deviations. The Shapiro–Wilk test was applied to assess data normality. When the null hypothesis was rejected, Wilcoxon’s signed-rank test was used, whereas Student’s *t*-test was employed when normality was confirmed. All analyses were two-tailed, with statistical significance defined as *p* < 0.05. To establish the relevant cutoff values (CO) for IBMP-ELISA, ten *T. cruzi*-reactive and ten *T. cruzi*-nonreactive samples were assessed simultaneously in all microplates. These samples had previously been characterized as positive or negative based on two serological tests following international guidelines [37,38]. CO values were derived by calculating the largest area under the ROC curve, which defined the maximum optical density (OD) required to discriminate between reactive and nonreactive *T. cruzi* samples. The results were expressed as a reactivity index (RI), representing the ratio of sample OD to CO OD. RI values > 1.00 were considered positive, while samples with RI values falling within the indeterminate zone (RI values of 1.0 ± 10%) were classified as inconclusive. A flow chart (Figure 1) was constructed in accordance with the Standards for Reporting of Diagnostic Accuracy Studies (STARD) guidelines [39].

## 3. Results

A total of ten previously collected, anonymized human serum samples were included in this study. The study sample had a mean age of 12.5 years (interquartile range [IQR]: 2.8–25.8 years), with a female-to-male ratio of 0.43:1. Among individuals exclusively infected with *Crithidia* sp. LVH-60A, the mean age was 10 years (IQR: 3.0–25.0 years) with a female-to-male ratio of 1.33:1. For the three individuals tested positive for *L. infantum*, the mean age was 19.0 years (IQR: 4.0–48.0 years), and no females were present. The majority of individuals reside in the Sergipe state (Figure 1). Two individuals live in the state of Bahia but seek medical care in Aracaju, the capital of the state of Sergipe, due to proximity. No information regarding the city of origin was available for a male child aged two years and infected with *L. infantum*.

Among the ten serum samples, latent class analysis revealed that two *Crithidia* sp. LVH-60A-positive samples (28.6%) were potentially coinfected with *T. cruzi*. As a result, these two *T. cruzi*-positive samples were excluded from the study (Figure 2). According to the LCA classification, both samples were assigned as P5, indicating positivity for all four IBMP proteins with a posteriori probability of 100%. One sample belonged to a 25-year-old man residing in Estância-SE, while the other sample was obtained from a three-year-old boy living in Olindina-BA. Information regarding the likely routes of *T. cruzi* infection was not available. The remaining eight samples were classified as *T. cruzi*-negative, with seven samples classified as P1 (negative result for all four IBMP proteins) and one sample classified as P2 (negative result for three IBMP proteins). Notably, all *L. infantum*-positive samples tested negative for *T. cruzi*.

Following the serological definition of the samples as *T. cruzi*-positive or *T. cruzi*-negative using LCA, we investigated the potential antigenic cross-reactivity of four IBMP chimeric *T. cruzi* antigens with human leishmaniasis and *Crithidia* sp. LVH-60A antibodies (RI > 1.10) via ELISA. As illustrated in Figure 3, the chimeric antigens exhibited minimal cross-reactivity. Both *Crithidia* sp. LVH-60A (0.42–0.55)- and *L. infantum*-positive samples (0.48–0.55) exhibited low mean RI values across all four chimeric antigens. Notably, none of the analyzed antigens showed cross-reactions in the *Crithidia* sp. LVH-60A-positive samples. Similar results were noted for the *L. infantum*-positive samples, except for IBMP-8.1, which yielded a positive result in one sample (RI = 1.45). Undesirable results (gray zone) were observed in one sample when tested with IBMP-8.2 and IBMP-8.3, while two samples showed undesirable results with IBMP-8.4. All samples with these characteristics presented RIs ranging from 0.90 to 1.00. Regarding *L. infantum*, only one sample fell within the gray zone when tested with IBMP-8.4 (RI = 0.94).

## 4. Discussion

Accurate and specific diagnostic methods are crucial for effectively managing and controlling infectious diseases. Chagas disease (CD) and leishmaniasis are significant neglected tropical diseases that often coexist in overlapping geographic regions. However, diagnosing these diseases can be challenging due to potential cross-reactivity in serologic testing. Cross-reactivity occurs when antibodies produced in response to one pathogen mistakenly recognize antigens from another, resulting in false-positive or inconclusive results. The challenges in diagnosing CD are further amplified when considering *Trypanosoma rangeli* [25,40,41], *Trypanosoma evansi* [42], and other members of the Trypanosomatidae family, including *Crithidia*, *Herpetomonas*, *Blastocrithidia*, and *Leptomonas genera*, as well as newly discovered species like *Crithidia* sp. LVH-60A [34]. However, due to the difficulty of identifying individuals infected with these species, there is a lack of studies investigating their influence on the serological diagnosis of CD. To address this knowledge gap, our study investigated the influence of anti-*Crithidia* sp. LVH-60A and anti-*L. infantum* antibodies in the diagnosis of CD, utilizing four IBMP *T. cruzi* chimeric proteins as antigens.

Cross-reactivity between anti-*Leishmania* spp. antibodies and commercial tests used for diagnosing *T. cruzi* infection has been extensively reported [13,22,25,32,43,44,45,46]. In a specific study, the use of epimastigote extracts from the Ninoa and Queretaro strains resulted in 16% cross-reactivity in sera from individuals infected with *Leishmania* spp. [45]. Another study focused on epimastigotes of the DTU TcI Dm28c *T. cruzi* strain and observed 33% cross-reactivity in sera from individuals with visceral leishmaniasis [46]. Similarly, employing a total extract of epimastigotes from the *T. cruzi* Y strain as an antigen revealed cross-reactivity rates of 92.31% and 75% in sera from individuals with visceral and American cutaneous leishmaniasis, respectively [25]. The same authors also reported high seropositivity when evaluating the performance of commercial ELISA kits using *T. cruzi* epimastigote antigens. Our group also reported cross-reactivity rates ranging from 19.3% to 54.8% and from 18% to 20.9% for two conventional tests used in diagnosing CD in samples positive for American cutaneous leishmaniasis (*n* = 572) and visceral leishmaniasis (*n* = 172), respectively [22]. However, when evaluating cross-reactivity of IBMP antigens using the same panel, significantly lower rates were observed. For American cutaneous leishmaniasis, cross-reactivity rates ranged from 0.35% (IBMP-8.3) to 0.70% (IBMP-8.1 and IBMP-8.2). No cross-reactivity was detected for IBMP-8.4. Regarding visceral leishmaniasis, cross-reactivity was only observed for IBMP-8.2 (3.49%) and IBMP-8.3 (0.58%) [22].

In our present study, we did not observe any cross-reactivity for *L. infantum*, except for a single positive sample that tested positive with IBMP-8.1. This positive sample was obtained from a 48-year-old male from Canindé do São Francisco. The reactivity index for IBMP-8.1 was 1.45, while for other molecules, it ranged from 0.81 (IBMP-8.2) to 0.88 (IBMP-8.3) and 0.94 (IBMP-8.4). Despite the negative outcomes for antigens IBMP-8.2, IBMP-8.3, and IBMP-8.4, their proximity to the cutoff (RI = 1.00) suggests the possibility of a positive status for CD in this patient. However, it is plausible that these specific results represent false negatives for these samples, thus excluding cross-reaction. Therefore, no cross-reaction would be considered. To address this uncertainty, further analysis using new samples or employing high analytical sensitivity tests, such as chemiluminescence or liquid microarray assay, could provide a conclusive answer.

Serologic cross-reactivity among different infectious agents is extensively documented in the literature. In the case of human *T. cruzi* infection, cross-reactivity has been reported with a range of pathogens, including toxoplasmosis, dengue, filariasis, schistosomiasis, measles, rubella, HIV-1/2, HTLV-1/2, HBV, HCV, syphilis, Zika virus, Hanseniasis, tuberculosis, autoimmune diseases, and leishmaniasis, as previously reported [25,32,40,42]. Few investigations have reported cross-reactivity for *T. rangeli* [25,40,41] and *T. evansi* [42], while no studies have explored cross-reactivity among *Crithidia*, *Herpetomonas*, *Blastocrithidia*, *Leptomonas*, or *Crithidia* sp. LVH-60A. Notably, *Crithidia* sp. LVH-60A is a recently discovered species in the Brazilian state of Sergipe [34], and no prior studies have examined the potential impact of anti-*Crithidia* sp. LVH-60A antibodies on generating false-positive or inconclusive results in CD serological testing. Therefore, our study represents the first investigation to address this research gap by using four *T. cruzi* chimeric antigens for CD diagnosis. Importantly, no cross-reaction was observed. Two *Crithidia* sp. LVH-60A-positive samples tested positive for all four IBMP antigens, indicating coinfection with *T. cruzi*. These samples were obtained from individuals residing in Olindina (Bahia) and Estância (Sergipe), both cities located in a CD endemic area.

Understanding cross-reactions in serological tests for CD is vital for several reasons. Initially, it plays a crucial role in distinguishing CD from other conditions with similar clinical symptoms, such as leishmaniasis or other Trypanosomatid parasites. False positives have significant implications beyond diagnostic inconvenience; they can lead to unnecessary treatments and emotional distress for patients. Recognizing and accounting for cross-reactions enables more precise and reliable diagnoses. Additionally, a thorough understanding of the scope and mechanisms of cross-reactions provides valuable insights into the epidemiology of CD and related diseases. This understanding empowers researchers and healthcare professionals to monitor the prevalence of CD and concurrent infections, facilitating a holistic approach to disease management. In regions where CD coexists with other parasitic diseases, the ability to differentiate between these conditions is of paramount importance for effective disease control strategies. Being vigilant about recognizing cross-reactions guides the development of targeted public health measures, optimizes resource allocation, and reduces the potential for misdirected interventions. Identifying the origins of cross-reactions is a catalyst for developing more specific diagnostic tests, particularly in regions with multiple coexisting infections, contributing to the continuous improvement of diagnostic tools.

A main limitation of our study was the small number of *Crithidia* sp. LVH-60A-positive samples analyzed. However, despite this limited sample size, the significance and novelty of the data presented here are evident. Recognizing and understanding cross-reactions in serological tests for CD are essential for ensuring accurate diagnoses, enhancing patient care, and bolstering the effectiveness of CD and related disease control efforts. Furthermore, this knowledge supports ongoing refinement of diagnostic tools, ultimately improving disease management, enhancing epidemiological surveillance, and advancing diagnostic methodologies. To the best of our knowledge, no previous investigations have attempted to assess the cross-reactivity of anti-*Crithidia* sp. LVH-60A antibodies in CD serology, making our findings particularly noteworthy. Importantly, no cross-reaction was observed. In summary, our findings suggest that the use of IBMP chimeric antigens holds promise for the safe diagnosis of CD in regions where *Crithidia* sp. LVH-60A and *T. cruzi* coexist. Nevertheless, further investigations involving a larger number of *Crithidia* sp. LVH-60A-positive samples are required to confirm our findings.

## Figures and Tables

**Figure 1 diagnostics-13-03470-f001:**
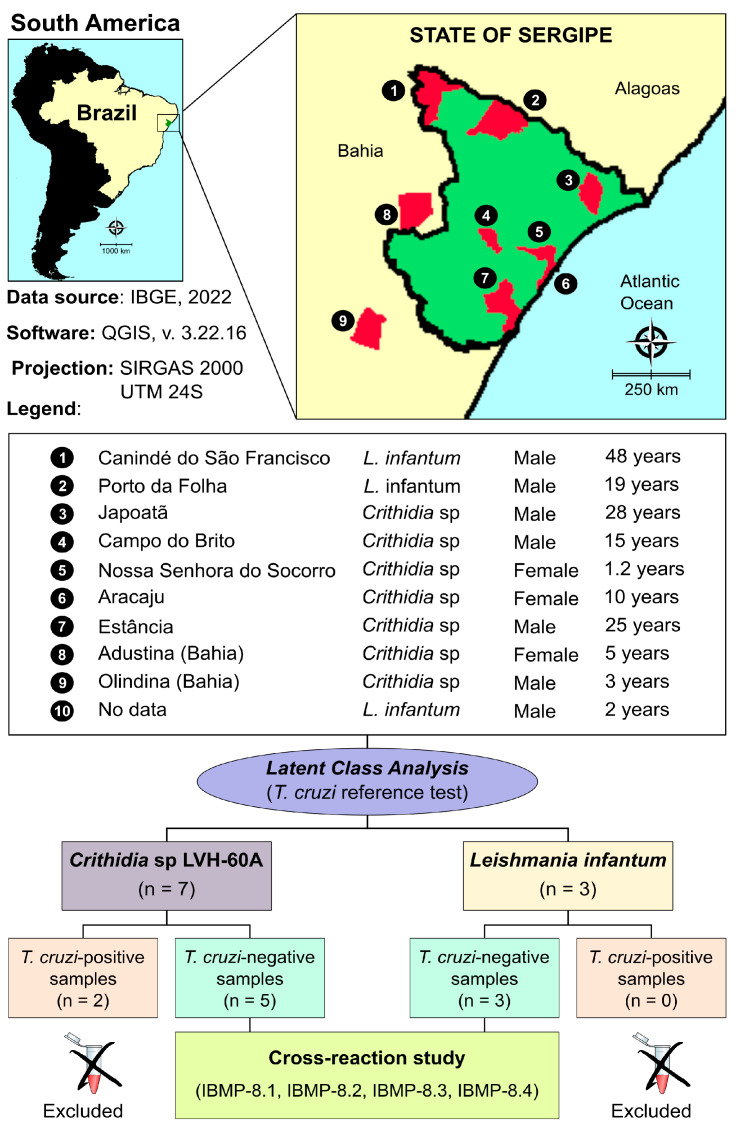
Flowchart illustrating study design in conformity with the Standards for Reporting of Diagnostic Accuracy Studies (STARD) guidelines. Public domain digital map was freely obtained from the Brazilian Institute of Geography and Statistics (IBGE) cartographic database in shapefile format (.shp), which was subsequently reformatted and analyzed using QGIS version 3.22.16 (Geographic Information System, Open Source Geospatial Foundation Project. http://qgis.osgeo.org accessed on 27 September 2023).

**Figure 2 diagnostics-13-03470-f002:**
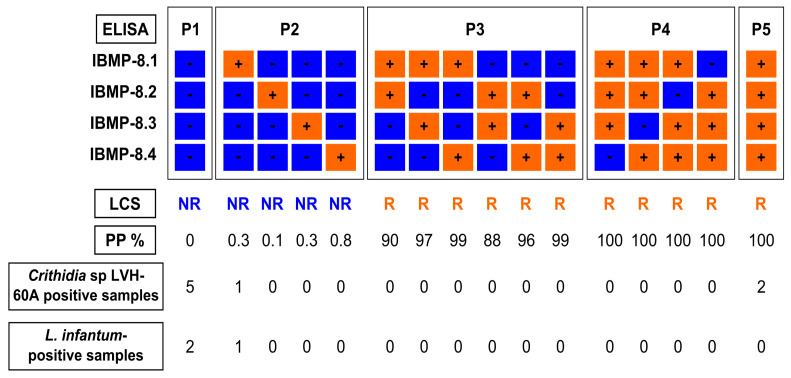
Reaction patterns of chimeric antigens in latent class analysis (LCA) used in anti-*T. cruzi* ELISA tests. LCS, latent class status; NR, nonreactive; PP, a posteriori probability; R, reactive; P1, P2, P3, P4, and P5, reaction response; N, number of samples.

**Figure 3 diagnostics-13-03470-f003:**
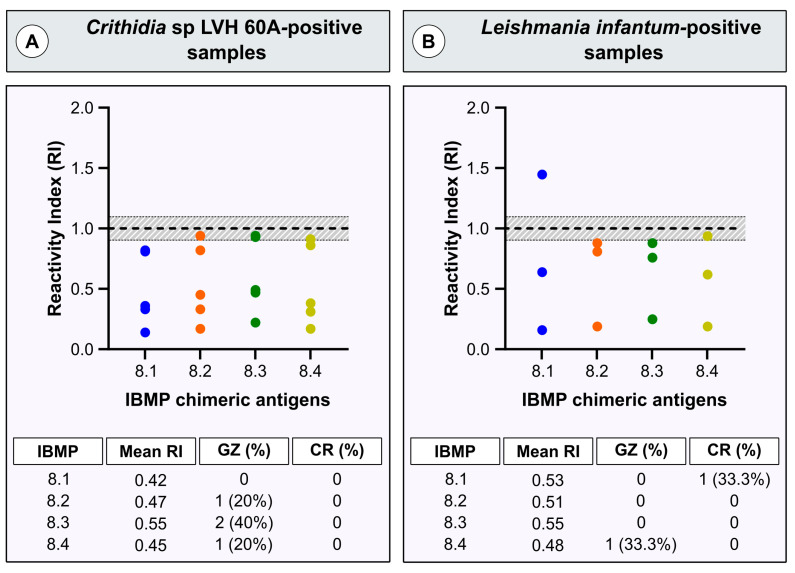
Reactivity index values from *Crithidia* sp. LVH-60A- and *Leishmania infantum*-positive serum samples assayed with four IBMP chimeric antigens. RI = 1.0, cutoff; RI = 1.0 ± 10% (shaded area), gray zone. Horizontal lines for each group of results: geometric means; GZ, gray zone; CR, cross-reactivity; RI, reactivity index.

## Data Availability

The raw data supporting the conclusions of this article are available from the authors without reservation.

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
