# Peer review of "Assessment of Cross-Reactivity of Chimeric Trypanosoma cruzi Antigens with Crithidia sp. LVH-60A: Implications for Accurate Diagnostics"

_diagnostics, 2023, doi:10.3390/diagnostics13223470_

Round 1
Reviewer 1 Report
Comments and Suggestions for Authors
The manuscript "Assessment of Cross-Reactivity of Chimeric Trypanosoma cruzi Antigens with Crithidia sp LVH-60A: Implications for Accurate Diagnostics" is well written. Although the sample size was small, the results obtained are promising for further research. I recommend accepting this work with minor changes and suggest that the authors address the following points:
1) In line 65-66, the references to Western blot [7] and lateral flow assays [7] do not provide any information about these analytical methods in the referenced work. It is necessary to include links to the original articles describing the development of these tests, or if they are commercial test systems, provide links to the company's website.
2) In line 149, there seems to be a typo regarding "infection. his well-established." Please clarify.
3) In the figure 2, class P2. The absence of cross-reactivity with IBMP-8.1 in the second sample is indicated in orange. Does this mean anything?
Author Response
We extend our gratitude to both you and the reviewer for your insightful recommendations, which have significantly enhanced our manuscript. We believe that this revised version now offers a more refined and equitable depiction of our research. We are optimistic that the manuscript aligns well with the standards of your esteemed journal and kindly request your consideration for publication. The responses to the questions have been provided below for your reference (Manuscript ID: diagnostics-2662007).
1) In line 65-66, the references to Western blot [7] and lateral flow assays [7] do not provide any information about these analytical methods in the referenced work. It is necessary to include links to the original articles describing the development of these tests, or if they are commercial test systems, provide links to the company's website.
Reply: We appreciate the reviewer's suggestion, and we have made the changes as follows:
For Western blot:
Before: Gomes, Y.M.; Lorena, V.M.B.; Luquetti, A.O. Diagnosis of Chagas Disease: What Has Been Achieved? What Remains to Be Done with Regard to Diagnosis and Follow up Studies? Mem Inst Oswaldo Cruz 2009, 104 Suppl, 115–121, doi:10.1590/S0074-02762009000900017.
After (lines 421-424): Daltro, R.T.; Santos, E.F.; Silva, Â.A.O.; Maron Freitas, N.E.; Leony, L.M.; Vasconcelos, L.C.M.; Luquetti, A.O.; Celedon, P.A.F.; Zanchin, N.I.T.; Regis-Silva, C.G.; et al. Western Blot Using Trypanosoma cruzi Chimeric Recombinant Proteins for the Serodiagnosis of Chronic Chagas Disease: A Proof-of-Concept Study. PLoS Negl Trop Dis 2022, 16, e0010944, doi:10.1371/journal.pntd.0010944.
For lateral flow:
Before: Dias, J.C.P.; Ramos Jr., A.N.; Gontijo, E.D.; Luquetti, A.; Shikanai-Yasuda, M.A.; Coura, J.R.; Torres, R.M.; Melo, J.R. da C.; Almeida, E.A. de; Oliveira Jr., W. de; et al. 2 nd Brazilian Consensus on Chagas Disease, 2015. Rev Soc Bras Med Trop 2016, 49, 3–60, doi:10.1590/0037-8682-0505-2016.
After (lines 425-427): Silva, E.D.; Silva, Â.A.O.; Santos, E.F.; Leony, L.M.; Freitas, N.E.M.; Daltro, R.T.; Ferreira, A.G.P.; Diniz, R.L.; Bernardo, A.R.; Luquetti, A.O.; et al. Development of a New Lateral Flow Assay Based on IBMP-8.1 and IBMP-8.4 Chimeric Antigens to Diagnose Chagas Disease. Biomed Res Int 2020, 2020, 1803515, doi:10.1155/2020/1803515.
2) In line 149, there seems to be a typo regarding "infection. his well-established." Please clarify.
Reply: We appreciate the reviewer for bringing this to our attention, and we have revised the sentence as follows:
Before: “…infection. his well-established and validated approach [23] determined…”
After (lines 183): “…infection. This well-established and validated approach [23] determined…”
3) In the figure 2, class P2. The absence of cross-reactivity with IBMP-8.1 in the second sample is indicated in orange. Does this mean anything?
Reply: We appreciate the reviewer for pointing out this issue, which pertains to an error in the figure’s coloring. We have adjusted the color accordingly. Please see Figure 2.
Reviewer 2 Report
Comments and Suggestions for Authors
Dear authors, congratulations for your work!
The paper comes to face a major health problem in Brasil. The results obtained by the research team suggest that IBMP antigens offer promising diagnostic performance. But the number of positive subjects is low. In this regard, the study must be continued on a large number of patients.
Also, more references should be added.
Author Response
We extend our gratitude to both you and the editor for your insightful recommendations, which have significantly enhanced our manuscript. We believe that this revised version now offers a more refined and equitable depiction of our research. We are optimistic that the manuscript aligns well with the standards of your esteemed journal and kindly request your consideration for publication. The responses to the questions have been provided below for your reference (Manuscript ID: diagnostics-2662007).
Dear authors, congratulations for your work! The paper comes to face a major health problem in Brazil. The results obtained by the research team suggest that IBMP antigens offer promising diagnostic performance. But the number of positive subjects is low. In this regard, the study must be continued on a large number of patients. Also, more references should be added.
Reply: We appreciate the reviewer's encouragement. We plan to increase the number of samples in the future, but this could take years due to the rarity of human Crithidia infection. Despite the potential for a lengthy process, we will retain all serum samples identified as Crithidia-positive. In response to the reviewer's suggestions, we have incorporated additional references, as highlighted below:
References:
Lines 395-396: López-García, A.; Gilabert, J.A. Oral Transmission of Chagas Disease from a One Health Approach: A Systematic Review. Tropical Medicine and International Health 2023, 28, 689–698.
Lines 397-398: Chancey, R.J.; Edwards, M.S.; Montgomery, S.P. Congenital Chagas Disease. Pediatr Rev 2023, 44, 213–221, doi:10.1542/pir.2022-005857.
Lines 399-400: Wendel, S. Transfusion Transmitted Chagas Disease: Is It Really under Control? Acta Trop 2010, 115, 28–34, doi:10.1016/j.actatropica.2009.12.006.
Lines 401-402: Radisic, M. V.; Repetto, S.A. American Trypanosomiasis (Chagas Disease) in Solid Organ Transplantation. Transplant Infectious Disease 2020, 22, doi:10.1111/tid.13429.
Lines 403-404: Hofflin, J.M.; Sadler, R.H.; Araujo, F.G.; Page, W.E.; Remington, J.S. Laboratory-Acquired Chagas Disease. Trans R Soc Trop Med Hyg 1987, 81, 437–440, doi:10.1016/0035-9203(87)90162-3.
Lines 421-424: Daltro, R.T.; Santos, E.F.; Silva, Â.A.O.; Maron Freitas, N.E.; Leony, L.M.; Vasconcelos, L.C.M.; Luquetti, A.O.; Celedon, P.A.F.; Zanchin, N.I.T.; Regis-Silva, C.G.; et al. Western Blot Using Trypanosoma cruzi Chimeric Recombinant Proteins for the Serodiagnosis of Chronic Chagas Disease: A Proof-of-Concept Study. PLoS Negl Trop Dis 2022, 16, e0010944, doi:10.1371/journal.pntd.0010944.
Lines 425-427: Silva, E.D.; Silva, Â.A.O.; Santos, E.F.; Leony, L.M.; Freitas, N.E.M.; Daltro, R.T.; Ferreira, A.G.P.; Diniz, R.L.; Bernardo, A.R.; Luquetti, A.O.; et al. Development of a New Lateral Flow Assay Based on IBMP-8.1 and IBMP-8.4 Chimeric Antigens to Diagnose Chagas Disease. Biomed Res Int 2020, 2020, 1803515, doi:10.1155/2020/1803515.
Reviewer 3 Report
Comments and Suggestions for Authors
Overall the manuscript has a number of good points. The limited scope of human subjects with verified infections by Crithidia is the greatest limitation. I feel that once edited for readability and grammar the contents are valid for this level of initial immunologic analysis. At this point the serologic data is not strong enough for human diagnostic use but it would be a first step to further applications of the tools discussed.
This manuscript could use overall grammar and spell checking. Examples below are among the many.
Line 54: Bugs do not have “Urine” “or urine of bloodsucking triatomines”
Line 72 and elsewhere: Leishmania sp. should be Leishmania spp.
Line 90 and Line 239: Trypanosomatidae family is redundant simply write Trypanosomatidae
Line 90-91: Leptomonas genera delete the “genera” as this is implied
Line 149: “infection. his well” is clearly a typographic error
Comments on the Quality of English Language
Overall the manuscript has a number of good points. The limited scope of human subjects with verified infections by Crithidia is the greatest limitation. I feel that once edited for readability and grammar the contents are valid for this level of initial immunologic analysis. At this point the serologic data is not strong enough for human diagnostic use but it would be a first step to further applications of the tools discussed.
This manuscript could use overall grammar and spell checking. Examples below are among the many.
Line 54: Bugs do not have “Urine” “or urine of bloodsucking triatomines”
Line 72 and elsewhere: Leishmania sp. should be Leishmania spp.
Line 90 and Line 239: Trypanosomatidae family is redundant simply write Trypanosomatidae
Line 90-91: Leptomonas genera delete the “genera” as this is implied
Line 149: “infection. his well” is clearly a typographic error
Author Response
We extend our gratitude to both you and the editor for your insightful recommendations, which have significantly enhanced our manuscript. We believe that this revised version now offers a more refined and equitable depiction of our research. We are optimistic that the manuscript aligns well with the standards of your esteemed journal and kindly request your consideration for publication. The responses to the questions have been provided below for your reference (Manuscript ID: diagnostics-2662007).
Line 54: Bugs do not have “Urine” “or urine of bloodsucking triatomines”
Reply: We appreciate the reviewer's suggestion, and we have revised the sentence accordingly:
Before: “…primarily occurs through the feces or urine of bloodsucking triatomines…”
After (lines 59-60): “…primarily occurs through the feces or urine of bloodsucking triatomines…”
Line 72 and elsewhere: Leishmania sp. should be Leishmania spp.
Reply: We appreciate the reviewer for bringing this to our attention, and we have made the replacement of “Leishmania sp.” with “Leishmania spp.” as follows:
Before: “…cross-reactivity with Leishmania sp. and Trypanosoma rangeli…”
After (line 81): “…cross-reactivity with Leishmania spp. and Trypanosoma rangeli…”
Before: “…Leishmania sp. has been consistently observed…”
After (line 85): “…Leishmania spp. has been consistently observed…”
Before: “…individuals exposed to Leishmania sp., especially …”
After (line 88): “…individuals exposed to Leishmania spp., especially…”
Before: “…Leishmania sp. coexist. In addition to Leishmania sp., other…”
After (line 120): “…Leishmania spp. coexist. In addition to Leishmania spp., other…”
Before: “…Cross-reactivity between anti-Leishmania sp. antibodies…”
After (line 279): “…Cross-reactivity between anti-Leishmania spp. antibodies…”
Before: “…from individuals infected with Leishmania sp. [35]…”
After (line 282): “…from individuals infected with Leishmania spp. [35]…”
Line 90 and Line 239: Trypanosomatidae family is redundant simply write Trypanosomatidae
Reply: We have modified the sentence as suggested by the reviewer:
Before: “…Trypanosomatidae family, including Crithidia …”
After (line 121): “…Trypanosomatidae family, including Crithidia …”
Before: “…the Trypanosomatidae family, including…”
After (line 272): “…the Trypanosomatidae family, including…”
Line 90-91: Leptomonas genera delete the “genera” as this is implied
Reply: We appreciate the reviewer's efforts to make the text more comprehensive. We have removed "genera," as follows:
Before: “…or Leptomonas genera, may also be present in…”
After (line 122): “…or Leptomonas genera, may also be present in…”
Line 149: “infection. his well” is clearly a typographic error
Reply: We appreciate the reviewer for bringing this to our attention, and we have revised the sentence as follows:
Before: “…infection. his well-established and validated approach [23] determined…”
After (line 183): “…infection. This well-established and validated approach [23] determined…”